# The liminal space between hope and grief: The phenomenon of uncertainty as experienced by people living with relapsing-remitting multiple sclerosis

Eva C. van Reenen[1]*, Inge A. M. van Nistelrooij[1], Leo H. Visser[1,2], Janet W. K. de Beukelaar[3], Stephan T. F. M. Frequin[4], Alistair R. Niemeijer[1]

1 Department of Care Ethics, University of Humanistic Studies, Utrecht, The Netherlands, 2 Department of Neurology, Elisabeth-TweeSteden Hospital, Tilburg, The Netherlands, 3 Department of Neurology, Albert Schweitzer Hospital, Dordrecht, The Netherlands, 4 Department of Neurology, St. Antonius Hospital, Utrecht, The Netherlands

☯ These authors contributed equally to this work.
* e.vanreenen@uvh.nl

## Abstract

### Background

People with the chronic disease Multiple Sclerosis are subjected to different degrees of profound uncertainty. Uncertainty has been linked to adverse psychological effects such as feelings of heightened vulnerability, avoidance of decision-making, fear, worry, anxiety disorders, and even depression. Research into Multiple Sclerosis has a predominant focus on the scientific, practical, and psychosocial issues of uncertainty. In comparison, existential uncertainty has been under-researched, even though it might pose a greater burden to those experiencing it.

### Objective

To understand the lived experience of uncertainty for people living with relapsing-remitting Multiple Sclerosis.

### Methods

This study followed a phenomenological research design, employing elements of both the Reflective Lifeworld Approach and Phenomenology of Practice. Seventeen people with a recent (<1 year) diagnosis of relapsing-remitting Multiple Sclerosis were included. In-depth interviews were conducted immediately after inclusion.

### Results

The lived experience of uncertainty can be described as a stumbling motion across the liminal space between hope and grief while dealing with oscillating feelings of unrest concerning the body, self, and others. The following four constituents further illuminate the meaning of the phenomenon: Having to constantly (re)define unfamiliar and intangible bodily changes

**Data Availability Statement:** The dataset is available in the DANS repository: https://doi.org/10.17026/SS/7T9L36.

**Funding:** The first author was generously funded by the Dutch National MS Foundation (https://www.nationaalmsfonds.nl/; grant number OZ2019-008) and by an Educational Grant (Uncertainty MS fellowship) from Merck Group (https://www.merckgroup.com/nl-nl/company.html). The funders had no role in study design, data collection and analysis, decision to publish, or preparation of the manuscript.

**Competing interests:** The authors have declared that no competing interests exist.

on one's own; Unsteady navigating amidst a destabilization of the imagined life; Relating to others as a source of, mirror or buffer for uncertainty; Going through overwhelming fears and worries while clinging to one's own logic.

## Conclusion

Adding to existing qualitative and phenomenological research into MS and theories on uncertainty, this study portrays uncertainty as a multifaceted experience. The findings imply the need for a continuous attunement of healthcare practitioners to the expectations, fears, avoidance techniques, and other uniquely personal circumstances of people with Multiple Sclerosis.

## Introduction

Multiple Sclerosis (MS) is a chronic disease of the central nervous system characterized by either transient or irreversible neurologic disability [1]. The prevalence figures of MS are rising, with over 2.8 million people living with MS worldwide [2, 3]. MS typically affects younger people (20–50 years) and at least twice as many women as men [2]. There are several clinical MS phenotypes, with relapsing-remitting MS (RRMS) being the most common one [4]. The disease course of the latter is characterized by alternating phases of attacks or relapses and complete or partial recovery [5, 6]. Due to the unpredictable nature of possible occurrence, timing, severity, location, and degree of recovery of symptoms and their treatment, people with MS are subjected to different degrees of (profound) uncertainty [7]. Some consider uncertainty to be one of the main challenges of living with the disease [8].

A systematic review of the qualitative literature examining the experiences of people with MS highlights the "inherent uncertainty people with MS experience[d] across all aspects of their lives" [9]. The uncertainty stems from not knowing diagnosis at first, clinical course, or future progression, and pertains to important life issues such as work, mobility, independence, and family life [9]. As reported by Nissen et al. [10], the experience of uncertainty is characterized by a lack of control and the attempt to reconcile varying symptoms and disease progression with personal needs. The extent to which an individual can tolerate uncertainty, has been linked to adverse psychological effects such as feelings of heightened vulnerability, avoidance of decision-making, fear, worry, anxiety disorders, and even depression [7, 11]. This implies that uncertainty can be associated with the well-being of those who experience it [7].

Uncertainty in healthcare was first theorized by Mishel [12]. Her Reconceptualized Uncertainty in Illness Theory (RUIT) was adjusted to apply to chronic illness or illness with the possibility of recurrence [13]. The model presents the antecedents of uncertainty, how people appraise the uncertainty (as either danger or opportunity), and what coping strategies they employ [14]. Although the work of Mishel is widely applied in practice and research, some claim it is one of many theories in a field of fragmented literature [15]. Han et al. define uncertainty at its core as "the subjective perception of ignorance" [15]. In an attempt to synthesize existing insights, Han and colleagues developed a 3-dimensional taxonomy that characterizes uncertainty through its sources (the probability, ambiguity, and complexity of a future event), issues, further divided into scientific (diagnosis, prognosis, and treatment recommendations), practical (system of healthcare), and personal (psychosocial and existential), and finally the locus of uncertainty (patient or healthcare provider) (Fig 1) [15].

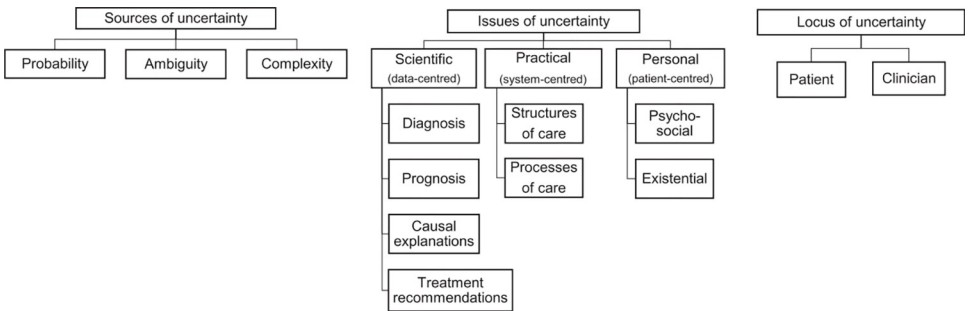

**Fig 1. The three dimensions of the Han et al taxonomy [16].**

Research into MS has a predominant focus on the scientific, practical, and psychosocial issues of uncertainty, e.g. the pathogenesis of MS and prognostic biomarkers [17], the efficiency of Disease Modifying Therapies (DMTs) in preventing relapses [18, 19], access to care [20, 21], mental health and coping [22–24], and measuring and targeting uncertainty and its repercussions [25, 26]. In comparison, existential uncertainty has not yet been explored, even though it might pose a greater burden to those experiencing it [16]. Existential uncertainty differs from other issues of uncertainty, as pertains to one's existence and is therefore both the source and object of uncertainty [16]. Dwan and Willig [16] point out that no amount of (medical) information could diminish existential uncertainty, as opposed to prognostic or scientific uncertainty. People can encounter health problems without experiencing uncertainty regarding diagnosis or prognosis, but will most likely always experience some form of personal uncertainty [16].

This gap mimics what Carel [27] describes as a distinction between disease and illness, objective body versus lived body, and biological dysfunction versus experience of illness. Healthcare focuses on biological dysfunction, its causes and treatments, and relies increasingly on medical technology for prevention, diagnosis, and treatment [27 p.15]. The illness experience, or "what it is like", is not accessible to the physician, other than through a second-person report [27 p.17/50]. This "epistemic restriction" may cause a physician to treat the disease without understanding the impact of the illness on a patient's life [27 p.50]. Insight into patients' experiences might improve healthcare practice for people with MS by closing the communicative and interpretative gap [27 p.51]. In 'Phenomenology of Illness', Carel [27 p.14] demonstrates how phenomenology is well-suited to analyzing the experience of illness as an existential transformation. Phenomenology sees people from a lifeworld perspective and provides insights into life "as we experience it" [28].

A phenomenological study of the lifeworld of people with MS is not a new venture. The phenomenological work of, amongst others, Toombs [29], Finlay [30], De Ceuninck van Capelle [31], and Van der Meide [32] have paved the way for the current study. However, the phenomenon of uncertainty within the lifeworld of people with MS, has never been the specific focus—one of the outcomes at most. The only exception, perhaps, is the article by Nissen and colleagues, who draw on the notion of "phenomenological uncertainty" in people with MS [10]. Interviews were, however, conducted as part of a larger anthropological project on care practices in everyday life with MS, and a thematic, instead of a phenomenological, analysis was performed. Furthermore, none of the participants had a recent (<1 year) diagnosis, while the early stages require great adjustments [23]. The aim of this study is therefore to understand the lived experience of uncertainty for people recently diagnosed with relapsing-remitting Multiple Sclerosis.

## Methods

### Study design

This study followed a phenomenological research design. Phenomenology turns to experiences of everyday life as we "live through" them and asks the question what it is like [33]. The study employed elements of the Reflective Lifeworld Approach (RLA) [34] and Phenomenology of Practice [35]. The RLA focuses on the lifeworld (lived and experienced world [34 p.125]) and aims to describe the "essence" of a phenomenon or "a structure of essential meanings" [34 p.245]. This essence is always bound to the lifeworld and therefore open and expandable; when the lifeworld changes, meaning changes as well [34 p.252]. In order for phenomena to present themselves as they are, not impose ourselves upon the things, and not make definite what is indefinite, an open (or phenomenological) attitude is required [34 p.121-122]. As researchers themselves exist in a meaningful lifeworld, it is impossible to achieve a blank slate [34 p.125]. One's preunderstandings can however be "bridled" [34 p.128]. With continuous reflection, a researcher critically considers (theoretical, scientific, cultural, or personal) preunderstandings, methods, and evolving meanings while increasing the validity of a research [34 p.163].

Although Dahlberg and Dahlberg [36] state that they never intended to present the RLA as a method with step-by-step instructions, the focus on parts of data and structuring them into meaning units and clusters of meaning [34], may imply that this process will, almost mathematically, lead up to an essence, leaving little room for the importance of writing [37]. Van Manen's Phenomenology of Practice therefore served as an inspiration to start writing, as writing is an indispensable part of phenomenological inquiry [35 p.365] and not a separate act of reporting on research findings [35 p.363-364]. It is through writing that meaning and insight emerge [35 p.367] while acknowledging the ambiguity of trying to put into words what cannot be put into words [35 p.244].

### Recruitment and participants

Participants were recruited as part of a larger prospective study through four Dutch hospitals with a specialized MS centre. The inclusion criteria for the larger study were a diagnosis of RRMS according to the McDonald criteria 2017 [38] and for the participant to be at least 18 years of age. Furthermore, participants needed to have a recent diagnosis of MS, meaning less than a year prior to inclusion, and be in the process of deciding on or starting with DMTs (which will be reported on in a separate article). At first, a call for participation was sent out to MS patients of one of the participating hospitals, and disseminated through the Dutch National MS Foundation. Although a large number of people responded (possibly indicating the relevance of the subject 'uncertainty'), none of these potential participants eventually fit the inclusion criteria. Second, a medical student, who worked as an intern on this research project in one of the participating hospitals, reached out to possible participants after their regular consultation with their physician. Simultaneously, neurologists and specialized MS-nurses working in the recruiting hospitals were instructed to approach patients who fit the inclusion criteria.

In phenomenological research, the size of the sample depends on the complexity of the phenomenon and is considered adequate when a sufficient rich description of the phenomenon has been obtained [34 p.175-176]. Seventeen participants were included in this study (Table 1) between 9 June 2021 and 21 April 2022. One of the participants (P12) was included with an initial diagnosis of RRMS, but their diagnosis was later changed to Clinically Isolated Syndrome (CIS). The research team decided to uphold the inclusion, for this diagnostic uncertainty seems especially relevant to people with (suspected) RRMS.

**Table 1. Participant characteristics.**

| Self-identified gender (n)[a] | Women 13 |
|---|---|
| | Men 4 |
| Age (years) | 25–60 (mean = 38) |
| Time between diagnosis and inclusion | 2 weeks—9 months (median = 8 weeks) |

[a] n = number of participants

## Data collection

In-depth interviews were carried out from June 2021 to April 2022, lasted between 39 and 120 minutes (median = 78 minutes), were audiotaped, and transcribed ad verbatim. Twelve interviews took place at participants' homes, an informal environment that is most likely to create a safe space for participants to elaborately share their experiences [35 p.315]. The other five were conducted in a private room at the hospital, usually before (N = 4) or after (N = 1) an appointment with a neurologist or specialized MS-nurse. All interviews were conducted by the first author.

Following a phenomenological research design, the interviews had an open character and were aimed at generating experiential data [34 p.184, 35 p.314]. An interview guide (S1 Appendix) was prepared in advance. The interviewer would start the conversation with an opening question regarding the subject of uncertainty and how this resonated with the participant. As the conversation unfolded, the interviewer would repeat certain words or phrases and encourage the participant to elaborate on actual events, to describe these events in as much detail as possible, and to refrain from abstract generalizations [34 p.184-186, 35 p.314-317, 39 p.199-201]. Follow-up questions were asked, such as "What do you mean by. . .?", "What happened when. . .?", "What is it like for you to. . .?", "What did you think, feel, and do at that particular moment?", and "Was there anyone with you and how?" [40]. Afterwards, the interviewer would capture non-verbal communication, atmosphere, and other observations in a journal, to encourage continuous reflection and increase the validity of the study [34 p.163].

## Data analysis

Although described here as two separate processes, data analysis and data collection are intertwined [41]. Analysis was performed by the first author and concurrently supported by and discussed with other members of the research team (IvN, AN, LV). Following the RLA, analysis was characterized by a movement between the whole, the parts, and the whole (or Gadamer's "hermeneutical rule") [34 p.236-237]. Initially, interview transcripts were read thoroughly and repeatedly. Preliminary thoughts, developing meanings, preunderstandings, and personal experiences were recorded in a reflexive journal and discussed in regular team meetings. Secondly, interview transcripts were divided into meaning units, reflecting the original text and participants' own words. The first author used Atlas.ti (version 23.1) as a tool to handle the large amount of data. Thirdly, clusters of meaning were formed. Although not customary to RLA, this was done for each participant individually, so as not to lose sight yet of the particular experiences [35 p.313].

At this point, the first author also started writing up experiential drafts of preliminary thematic understanding or constituents [34 p.255, 32 p.320]. These were discussed, compared, critically evaluated, and altered during frequent team meetings. It was crucial to maintain a bridled attitude in order to slow down, restrain preunderstanding, and refrain from forcing meanings to appear [34 p.242]. Reflexivity was practiced as a team activity and the discussions

between team members gradually gave shape to the essential meaning and its constituents, as will be described in the Results section.

## Ethical considerations

Prior to the start of this study, the Medical Ethical Review Committee Brabant (The Netherlands) established that the Dutch Medical Research Involving Human Subjects Act (WMO) did not apply (NW2020-65). In addition, the Ethical Review Committee of the University of Humanistic Studies Utrecht (The Netherlands) concluded that the rights, safety, and well-being of participants was guaranteed (2020.6). Furthermore, approval was obtained from the Review Boards of the four participating hospitals: the Elisabeth-TweeSteden Hospital Tilburg (L1135.2020), St. Antonius Hospital Utrecht (Z21.016), VU Medical Centre Amsterdam (2020.0715), and Albert Schweitzer Hospital Dordrecht (2021.044). Pseudonimity was ensured by assigning a number to all participants and removing as much identifying information from the transcripts as possible. Data were stored in a secured project file and access was restricted to members of the research team. All participants gave written informed consent.

The COVID-19 pandemic presented the researchers with ethical challenges. Visiting people with a chronic illness, often paired with a compromised immune system due to medication, at their home during a period of national lockdown, posed a risk to their health. Before making an appointment, the interviewer emphasized that all regulations (keeping 1,5m distance, wearing a mask, et cetera) would be complied with. An option to conduct the interview online was discussed as an alternative. It is worth noting that all participants agreed to a face-to-face interview, either at their own home or at the hospital.

## Results

*"It's really tough, losing that sense of control and power. I suppose control is always limited, one doesn't actually control what happens in life, but. . . now it feels like playing some kind of Russian roulette and you don't know how it's going to turn out." (P02)*

In this section, first the essence of the researched phenomenon is described, followed by its constituents. Based on the interviews and their subsequent analysis, the essence of the lived experience of uncertainty for people recently diagnosed with RRMS can be captured as *Stumbling across the liminal space between hope and grief while dealing with oscillating feelings of unrest towards body, self, and others*. If hope is one space and grief another, uncertainty seems to exist in a third in-between space. Being in uncertainty encompasses this space, an ambiguous desert island that is neither hope nor grief. The spaces exist in close proximity to one another. Hope is something to reach for, grief preferably shied away from or avoided altogether. Uncertainty is a restless state, lacking the steady promise of hope or the throbbing pain of grief. The feeling of unrest is fueled by the physical, mental, and social sequelae of the disease. Internal and external processes and events can expose ambivalence, but open endings are difficult to live with. The implications of certain happenings (are bended to) point towards enduring grief or fostering hope.

Four constituents further illuminate the meaning of the lived experience of uncertainty (Fig 2): *Having to constantly (re)define unfamiliar and intangible bodily changes on one's own; Unsteady navigating amidst a destabilization of the imagined life; Relating to others as a source of, mirror or buffer for uncertainty; Going through overwhelming fears and worries while clinging to one's own logic*. The constituents should not be viewed as separate entities but instead as overlapping elements of the same phenomenon. They will be described more elaborately

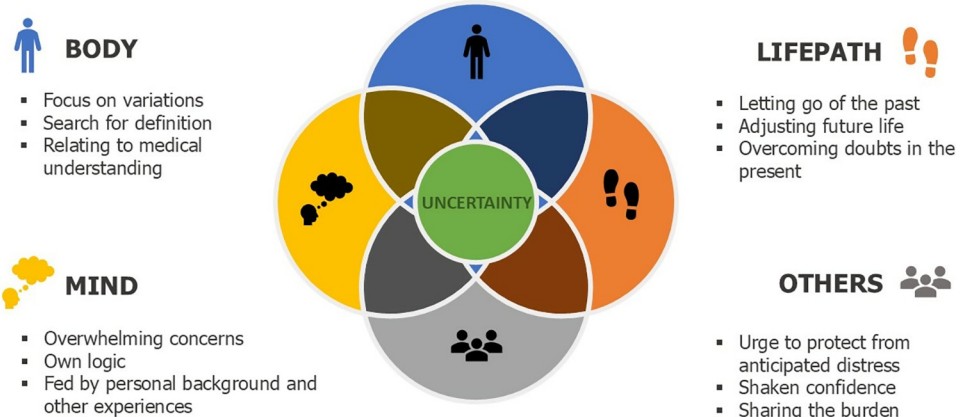

**Fig 2. The lived experience of uncertainty for people recently diagnosed with RRMS.**

below, starting each constituent with an extract from one of the interview transcripts and further supported by paraphrased examples from the different participants.

## Having to constantly (re)define unfamiliar and intangible bodily changes on one's own

> *"Sometimes, I get a little scared of a new exacerbation. Yesterday I felt something in my toe— which was probably because of a tight shoe—and when I became aware of this, I immediately started touching it. Does it tingle? Can I move it? So that anxiety is still in me, is still there." (P09)*

Participants are confronted with MS after experiencing alarming physical complaints, such as pain, muscle weakness, tingling in arms or legs, vertigo, or loss of sight, often completely out of the blue. It is as if their body abandoned them or at least caught them off guard (P02). The first few days or even weeks of persisting symptoms seem to revolve around surviving and finding a cause or explanation. After a while and in most cases aided by corticosteroids treatment, symptoms start to improve or disappear completely, making room for relief. There is, however, another challenge that remains: how to deal with an unfamiliar and unpredictable body with MS? Trust in and ability to interpret the body are fragile and need to be restored (P11, P16). This is a time consuming process and is aided by periods of relative stability.

**Focus on variations.** Amidst this regaining of trust in and familiarity with the body, an excessive focus on the body seems to take hold of the participants: a compelling need to establish what it can and cannot do and if any physical variation can be perceived. Even the slightest alteration could usher in a recurrence of previous symptoms or a new exacerbation (P09). It makes participants wonder whether their eyesight had always been a little blurry, or if it is a novel phenomenon caused by the MS (P06). Small and subtle somatic variations that might have gone unnoticed in the past now become the object of constant and fixated attention (P05). The diagnosis leads to questions and brings about fears and worries that were not there in the past and might be alleviated by answers and explanations. One's attention can, on the other hand, also be directed towards promising changes, suggesting an improvement in physical complaints (P13).

**Search for definition.** Some of the answers and explanations can be provided by a neurologist or specialized MS nurse. In their day-to-day lives, however, participants are mostly and

persistently navigating on their own. Constantly determining what they are and are not feeling. What is and what is not brought on by the MS. What they can and can no longer expect from their body. Although participants become increasingly knowledgeable of MS related phenomena, they lack a neurologist's medical expertise to interpret what is happening to them. They are acquainted with terminology such as exacerbation or relapse, but are not sure whether it is happening to them when they feel extra tired or experience more pain or trouble walking than usual (P03, P07, P13). They are aware of the fact that symptoms can improve over time and hope this is actually the case (P13, P16), but this is hard to determine (P08) and they might as well simply be getting accustomed to them (P11, P17). General fatigue complicates attuning to the signs of your body; is your past response to feeling tired still appropriate or how do you, under the current circumstances, know when to recharge your batteries (P05)? Even when you have a good day, you are still ill and need to take this into account (P06, P07). Taking care of yourself might mean taking a break where you might have carried on in the past (P13). How do you fill your days in a way that is not too strenuous if you keep guessing at what 'too strenuous' entails (P07)?

**Relating to medical understanding.**   Only sporadically do participants get the opportunity to have their bodies and its sensations observed and examined by a neurologist or specialized MS nurse. On a daily basis, they lack the objective measurements a healthcare practitioner can carry out and they need to rely on their own intuition. This is a challenge, especially as the value of this bodily know-how compared to medical understanding needs to be determined time and time again. Some put great trust in their own recognition of red flags and false alarms, for example when symptoms occur for the first time and participants just know something is off (P07, P15). They are persistent in their search for finding the right help, even if it is not immediately offered. Others start questioning their own perception because of external (medical) input (P12). What if you are told over and over again that your fatigue is probably stress-induced and that there is no definitive or objectifiable explanation for your loss of vision? Likewise, it is a confusing experience to learn that complaints you had in the past, might be related to MS in hindsight but never received such a serious label before (P13, P14, P15, P16). Medical proof seems to be valued more highly than participants' own feelings and judgements. In other cases, medicine lacks the tools to quantify particular phenomena such as fatigue or pain. Participants are encouraged to construe the implications and severity of these symptoms themselves. If you have experienced pain for years, even prior and unrelated to MS, and are fearful of complaining, you might be less likely to ask for pain medication (P03). Participants' attempt at (re)defining unfamiliar and intangible bodily changes is cluttered and not well defined.

## Unsteady navigating amidst a destabilization of the imagined life

*"This is something I really struggle with. At work they ask me what I think of it. My reply is 'I don't know'. 'What do you expect to be able to do?' 'I don't know.' 'Do you think you can work for two hours?' 'I DON'T KNOW!'. All I know, is that when I do something I'm very tired afterwards. I understand that I have to give in to my fatigue because I'll only make it worse if I don't. So now I have to figure everything out and think of the best way to go about it. I just don't know where to start rebuilding my life, experimenting with what I can and can not do, what I WANT especially since I may not be ably to do everything, energy wise." (P05)*

A possible or definitive MS diagnosis marks a moment of discontinuity in participants' lives as they had imagined it. The direction in which their path was heading, deliberately or

undeliberately, is no longer evident and may change completely. This change is hypothetical, since no-one can predict how, when, and even if it will occur (P08). Participants recognize that the future is uncertain for everyone, but they come to understand it is a different experience altogether with MS dangling over their heads like the sword of Damocles (P05, P06) or a dark cloud casting shadows over their future (P08). Life seems to have lost a certain amount of taken-for-grantedness and carelessness. The actual but mainly the potential physical, mental, and energetic disabilities of MS, give rise to questions concerning the core of participants' existence. Who they were, are, and who they can still be. It turns their gaze at the past, present, and future.

**Letting go of the past.**   The past seems to serve as a painful memory of how life used to be but may never be again. A mirror for examining yourself and your old habits. How often do you look into this mirror or would you rather back away from it? Participants' initial course of action seems to be to return to the comforting familiarity of their life as it was before the diagnosis (P06), at times forgetting about the diagnosis or doubting its reality altogether (P16). Sooner or later, however, something has got to give. Resuming all daily activities such as work, meeting with friends and family, and managing a household may prove to be unrealistic under the current circumstances (P07). Conceding to this is a difficult task, especially since many participants are young (20–40 years old) and in the 'prime of their lives'. Some of them have just started a new job (P10), started their own family (P09) or are planning to (P04), while others have a large group of friends to keep up with (P05). Giving in to seemingly inevitable adaptations is further complicated by the fact that they are not definitively inevitable. Energy levels might return to how they used to be and other complaints might eventually disappear. It is definitely a hope that many participants cling to (P07, P13, P16, P17). Some participants, however, start to reflect on how busy and demanding their lives used to be, question whether this is a durable lifestyle (P16), and even hold themselves partially accountable for getting the disease in the first place (P13). They more easily make adjustments in their lives, as if having something or someone to blame instead of endlessly guessing at cause and effect, alleviates these concessions.

**Adjusting future life.**   However, if you will not or only partially regain your past life, what future life do you want to build for yourself? Who do you want to be? Who are you still able to be? To what extent is being a patient part of your identity? Participants do not seem to actively choose and work on a specific scenario, but they do play out different ones in their mind. These scenarios concern different issues, often matters that did not seem pressing before (P05) or would not be considered pressing to other, healthy individuals, such as becoming a parent (P04, P05, P06, P13), the stability of a relationship (P06), or the ability to work (P11). The open and uncertain future forces participants to think about their passions and dreams (P08) and alternative ways of fullfilling them (P15). They start to work on this with a coach (P07), take concrete steps with their employer to find different tasks (P09, P13), imagine themselves doing volunteer work (P06), especially in the unfortunate case of a pending dismissal (P10), or take pleasure in being a grandparent for their grandchildren (P15). The indeterminate character of these questions and doubts causes some participants to shy away from thinking about a 'limited' future (P14). Others do allow themselves to think and plan ahead, but no more than a year (P12, P17). In any case, participants are trying to come to terms with a reduced version of a future once imagined, not knowing what this version is going to look like.

**Overcoming doubts in the present.**   Amidst the cracks in past life and envisioned future, participants find themselves making practical choices and overcoming doubts in the present. On the one hand to keep alive the hope of regaining certain aspects of their old life without MS. On the other hand to rebuild a different, new life with MS. One of the choices almost every participant faces is whether or not to start taking medication. This choice is full of

ambiguities and raises many questions in itself. As there is no one right answer, choosing between the different types of medication is often a tedious job (P08, P13, P14, P16). The choice whether to start at all, however, seems a much easier one to make, despite having to consider the risk of dangerous adverse effects (P06, P13, P15) and then actually having to endure unpleasant side effects (P07, P05). This determination seems to be fueled by hope for a positive outcome in the long run and fear of the negative consequences of not using medication (P08, P13, P15). Doing nothing seems to give rise to more anxiety than doing something (P06, P16). The same sense of 'personal responsibility' extends to starting other treatments, such as physiotherapy (P07, P08) or a rehabilitation program (P03). A neurologist's advice is often quickly followed, even though the therapies can greatly impact participants' daily life (P03). The prospect of other medications to try, treatments to follow, or medical aids available, provide the participants with hope and certitude (P13, P16). Lastly, there are individual participants who would rather not use medication or start other treatments (P14, P17). Proof of or faith in a favorable (natural) course seems to dominate their thoughts.

### Relating to others as a source of, mirror or buffer for uncertainty

> *"I immediately said to my boyfriend: 'If you don't want to be with me anymore, or if our future is gone. . .' Those were the things going through my mind at the time. You just don't know, because you drag somebody else into it as well. It's his problem too, of course. So those were the first thoughts I had. 'Is my future gone?' 'Will he stay with me?'" (P06)*

Experiencing uncertainty not seldomly pertains to others and is shaped by changing social relations. Participants do not only consider the consequences of their disease on their own lives, but also on the lives of the people around them. Will your parents worry (even more) about you from now on (P12, P16)? Is your diagnosis upsetting your children (P11, P14, P16)? Does cancelling on social engagements with friends mean that they will be disappointed in you (P05, P07)?

**Urge to protect from anticipated distress.** Participants seem to feel an urge to protect them from any anticipated distress and share different amounts of (medical) information with different people. Some participants immediately call their parents after a hospital appointment, knowing that they are anxiously awaiting the results at home (P04, P06). Others want to keep up appearances with their parents and withhold the truth about how they are actually doing (P07) or try not to cry in front of their children (P16). Where children are concerned, participants often decide to share very little with them (P11). The thought of your children hurting on your account seems almost unbearable (P16). This attitude may even extend to children not yet born and reinforces the question of whether or not to have them (P04, P06, P12).

**Shaken confidence.** While this protective instinct is mostly connected to the effect participants expect their diagnosis to have on others, an opposite impact can also be noticed. What others think, say, or do can affect participants as well. It can draw their attention to sensitive topics, shake their confidence, or make them feel misunderstood. For instance, one participant feels like he is handling beginning cognitive impairments well, but his wife and children notice that he is often repeating himself and displaying different, inward behavior (P01). Through the eyes of his family, the participant notices a possible sign of disease progression he was not aware of or not yet ready to face. Likewise, the circumstances or decisions of other people, especially of those who also suffer from MS, can make participants question their own situation or course of action. Such as when they learn about cases with an adverse disease trajectory and fear this is what lies ahead for themselves (P11, P13). Furthermore, because it is often hard

to explain to others exactly what MS entails, participants are at risk of being exposed to inappropriate or even hurtful comments. From friends, who express feeling tired as well, when you try to explain your fatigue to them (P05). Or a family member who wonders whether your demanding job could have triggered the MS (P09). The unpredictable and at times unsupportive nature of other people's reactions triggers certain hesitancy in some participants to share their thoughts and feelings.

**Sharing the burden.**   However, others can also make participants' uncertainty more manageable by sharing the burden. A sister or girlfriend who encourages you to contact a doctor once more because you are clearly suffering, even though this did not amount to anything the first time (P04, P07). Family members who assure you not to worry about the diagnosis being a grim one (P13, P16). A husband who accompanies you to every hospital appointment (P16) and helps you decide on what medication to start (P14). Parents who provide you with questions to ask your neurologist (P12). Healthcare practitioners who go the extra mile to help you when there are no clear solutions or answers (P06, P07, P13, P16). Stories of other people with MS who are doing relatively well that lift your spirits (P11, P13, P16).

## Going through overwhelming fears and worries while clinging to one's own logic

*"I suppose I am the type of person who thinks about their future almost every day, almost every moment. What I might lose, what I might no longer be able to do, and what my life will look like. When I'm at work, for example, and busy repairing things, it's on my mind all the time. 'For how much longer will I be able to do this and... what should I do if I no longer can?' When I'm riding my motorbike, which is something I love to do, I'm also thinking 'For how much longer can I ride it? Should I sell it now or what should I do with it?' So during basically everything I do, including fun things, on holiday, I am constantly thinking about it."* (P04)

**Overwhelming concerns.**   The mind and body of participants are frequently occupied by perceptions, sensations, thoughts, and processes that relate to specific incidents, received information, decisions to be made, fears, or dreams. In such a moment, an immediate, primal, and uncontrollable response is induced; they are overwhelmed by strong emotions and pervasive questions. Questions that are often unanswerable but keep haunting them (P04, P14) and can be provoked by formerly routine and uneventful activities (P05). Taking the subway on a holiday may have been a quotidian task in the past but can now lead to worries about accessibility of public transport in the future (P06). Preparing dinner but not being able to cut the ingredients confronts you with the unsure prospect of improvement (P08). A visit to the hospital, especially in the period leading up to a diagnosis, can also result in anxious moments; being asked, for example, if a family member or partner is nearby, implying that bad news can be expected (P04). As though everything that is being said and done, loses its regular meaning and is ascribed an inflated one because of all the conceivable scenarios it paints.

**Own logic.**   As a means to counter or overcome these thoughts and feelings, and keep some hope alive, participants start conversing with themselves, talking in a strict and reprimanding way or calmly and comfortingly putting things into perspective. In doing so, participants seem to create their own logic that is not always logical, may contain inconsistencies, especially compared to the medical logic, and sometimes only partially acknowledges or completely evades certain facts, implications, and deductions. This own logic defies worries

(P08) and may postpone acceptance (P04), but runs the risk of denial (P14). The weight and complexity of certain information, however, can be too hard to grapple with.

Some participants, for example, assume that their initial complaints are not serious and will disappear on their own. Even if they are unfamiliar and alarming (P04, P15), and especially when healthcare practitioners are consulted who see no reason to investigate further (P12, P14). They come up with reasons that would explain tinglings, muscle weakness, or blurry eyesight, such as lying on a limb for too long at night (P07), needing new glasses (P16), engaging in home repair (P08), a recent pregnancy (P13), not getting enough exercise (P08, P13), or a stressful period at work (P09). Likewise, participants tell themselves there is no point in worrying about certain aspects of the future (P16), because no one knows what their future will look like (P06). They try to silence the worries by stressing that the possible scenarios may never play out (P08). Or they save up questions for their next appointment with their neurologist instead of looking into it right away (P06). In some cases, participants even actively avoid information about the disease, alternative treatments, and stories on the well-being of other MS patients, because they find them too confronting (P06, P07, P12, P14). Instead of gaining something from it, they fear it will only needlessly worry them (P16).

At other times, a participant might find it hard to come to terms with medical matters, such as a diagnosis or other (lack of) explanations. Not because they don't believe their doctor, but because the facts do not match their own logic (P08) or they are too vast to fathom (P14). For instance, when a participant keeps questioning the message that a cyst in her brain is benign and just an incidental finding, because an excruciating headache had been her first symptom of MS (P07). Some talk about their bodies as 'being able to tackle infections' (P11) or their MS as '(not) aggressive' (P02, P13) and cling to this reality without there being any explicit medical proof. Other participants keep requesting a new MRI scan sooner than is medically relevant (P01), repeatedly ask for an explanation they have already received (P03), or doubt the advice to start medical treatment as soon as possible (P17). Some participants want to explore other diagnoses, such as a Vitamin B12 deficiency (P11), an orthopaedic problem (P13), or consider pursuing a second opinion (P08, P17), before accepting they have MS. This seems paradoxical: by keeping alive the uncertainty about the diagnosis, they remain hopeful, yet restless at the same time.

**Fed by personal background and other experiences.**   All of these perceptions, thoughts, and processes are fed by the image of the disease, personal background, and other past and present experiences. Having been confronted in the past with severe cases of MS can lead to sombre ideas about the future (P15). One of the participants quickly jumps to anxious thoughts and sees this as a result of his father's terminal illness (P04). Another participant is reminded of a period in her life when she felt helpless and insecure due to several setbacks and is thrown into that same feeling of insecurity (P16). A general conception of healthcare practitioners being driven by money and the fear of legal allegations, can also instill a distrust in participants towards the medical advice they receive (P14). Others feel a strong affinity towards the medical profession because of family members who work in the field and put a great amount of trust in it (P08). One's own profession can similarly play a part. A participant who in the field of medical technology, relies heavily on the results of MRI scans (P01). Or the participant who works as a medical analist is constantly attempting to label everything, such as what subtype of MS she has or what qualifies as an exacerbation (P03).

## Discussion

The aim of this study was to better understand the phenomenon of uncertainty as it is experienced by people recently diagnosed with RRMS. The lived experience of uncertainty can be

described as a stumbling motion across the liminal space between hope and grief while dealing with oscillating feelings of unrest. This unrest concerns the body, self, and others. First, the findings will be reflected on in relation to existing literature. Second, the implications of the findings for healthcare practice will be discussed. Third, the limitations and strengths of the study will be reviewed.

## Reflections on the findings

The first constituent relates to the body and is characterized by an excessive focus on bodily changes, a rather lonely and continuous pursuit of clarification or explanations, and an attempt at defining the value of bodily know-how compared to medical understanding. Uncertainty pertaining to the body has been previously presented by Carel [27] as 'bodily doubt'. It is the breakdown of an unjustified and tacit sense of bodily trust in health towards a pervasive and core experience of illness [27 p.86-88]. Phenomenological features such as appearing suddenly, invading the normal sense of things, leaving a permanent mark, and revealing vulnerability and incapability all resonate with the findings of this study [27 p.93-95]. What this study adds is the recognition of the body as part of a complex, interwoven web of others, practices, and institutional systems [42]. The medical logic [43], discourse [44] or gaze [45] one is inevitably confronted with in illness, intensifies bodily doubt and covers it with extra layers of cognitive uncertainty. According to Van der Meide and colleagues [32], bodily uncertainty remains the hallmark of MS and is concerned with the body's unpredictability and (dis)ability in the present and less with (prognostic) uncertainty in the long-term. The current study connects bodily sensations to its implications for the future. In addition, Nissen and colleagues [10] similarly report on embodied fluctuations over time which require constant vigilance, learning about the body, and ongoing reconciliation with possible activities. There are, however, considerable differences in study design and execution. Most importantly, none of their participants had a recent (<1 year) diagnosis. The current study accentuates the first few months after the initial diagnosis of RRMS and distinguishes it as a period of heightened senses. In their 2016 article on patient perspectives on the process of MS diagnosis, De Ceuninck van Capelle and colleagues [46] describe this as an "increased awareness of the body" before, during, and shortly after receiving the diagnosis.

The second constituent relates to the self finding and coming to terms with a new path in life. The past figures as a (painful) memory of how life used to be but may never be again, and the present foregrounds choices that pave the way for a desired future life. Although the diagnosis as a breaking point and adapting to a life with MS has been widely described [9, 23, 30, 31, 47, 48], the uncertainty that permeates these experiences remains underexposed. Strickland, Worth, and Kennedy [49] conceptualize the diagnosis of MS as a "'threshold moment' where the individual's sense of self is disrupted from the former taken-for-granted way of being". The authors describe the time prior, during, and after diagnosis as a liminal state, a self in transition, "betwixt and between the old self and new self" [49]. Strickland and colleagues [49] acknowledge that the transition to a new self is subject to ongoing uncertainty that leads individuals to adopt a certain way of coping to minimise their discomfort [49]. Adding to the work of Strickland and colleagues, this study portrays different, overlapping layers of uncertainty, how participants try to make adjustments, the difficulties they face, and what this 'certain way of coping' looks like (see the fourth constituent), especially in that first period after receiving the diagnosis.

The third constituent has a distinctive social dimension. Others can heighten or relieve feelings of uncertainty. Not knowing what someone else is feeling or how someone else will respond, adds another layer to already existing uncertainties. At the same time, a loved one

can lift the solitary burden of uncertainty. Changes in family dynamics due to an MS diagnosis have been previously described [50] but generally, the relational world of people with MS has been subject of little research [51]. Parkinson et al. [51] performed a scoping review of the qualitative literature on the experiences of people living with a person with MS. Strikingly, "living with uncertainty" is one of the five themes identified [51]. Partners, children and parents of people with MS, and other carers experience the unpredictable trajectory of the disease as a constant source of worry and this can even lead to anxiety and depression [51]. The authors conclude that the experiences of the people who live with a person with MS closely resemble those of people with MS themselves (and are thus "intertwined like a double helix" as the title of their article suggests) [51]. Combining the results of this scoping review and the current study, shows that all parties experience uncertainty, not only due to the diagnosis but also due to the (heartbreaking) fact that they care for and about each other. This interconnectedness is, however, not represented in the design of these or other studies. To truly acknowledge our relationality would mean for future research to move away from the dichotomy between people with MS on one hand and people who live with a person with MS on the other [42].

The fourth and final constituent yields insight into the intricate mental structures evoked by uncertainty. What stands out, is that most participants come up with ways to fill in open endings. In moments of strong emotions or nagging worry, they create their own logic. Much has been written on how people with MS (attempt to) cope with their chronic illness [52–57]. In 2015, Alschuler and Beier [7] introduced the concept of intolerance of uncertainty to better understand how people with MS cope with its inherent uncertainty. Individuals who are most intolerant of uncertainty reside at one end of a continuum and are most likely to engage in thoughts and behaviors by which they attempt to exert control over the situation or eliminate a possibility [7]. They adopt coping strategies that have been associated with poorer psychological outcomes in terms of well-being and acceptance [7]. Following the work of Alschuler and Beier [7], the participants of this study seem to engage in negative coping behavior. However, framing intolerance of uncertainty as a static personality trait, does not seem to do justice to the varying nature and occurrence of certain thoughts, feelings, and (mental) responses over time. Hillen and colleagues [11] state that more research is needed to determine how differences in personalities and situational factors interact. In their 2017 article, Hillen et al. [11] present an integrative conceptual model of uncertainty as a broad phenomenon, both spatially and temporally, that encompasses different psychological dimensions, needs, motivations, and goals [11]. For example, uncertainty can evoke both cognitive, emotional, and behavioral responses. One of the cognitive responses is exemplified by the need for closure (NFC) in the face of uncertainty [11]. NFC causes "people to 'seize' on information that yields closure and to 'freeze' on closure once attained" [11]. This strongly resembles the experiences of our participants and there are other similarities between the current findings and the integrative model of uncertainty tolerance, such as its multidimensional character, evoking both positive and negative psychological responses, and deriving from both internal and external stimuli. Our empirical findings can enrich the theoretical model and vice versa, especially in light of the development of interventions aimed at (in)tolerance of uncertainty for people with MS [25, 58].

## Recommendations for healthcare practice

The findings imply the need for a continuous attunement of healthcare practitioners to the expectations, fears, avoidance techniques, and other uniquely personal circumstances of the people with MS in front of them [31]. Providing them with all available information might work for some individuals—although they should be cautious of the illusion of certainty and control [59]—but for others, a more cautious feeding of information is required. Issues

concerning family life cannot be resolved in a consultation room, but acknowledging the person with an illness as more than a diseased body, part of a complex, interwoven web of others, practices, and institutional systems, could help attune to a patient's needs [31, 60]. Studies show that there is a tendency in healthcare to focus on scientific uncertainty (about diagnoses, prognoses, causes, and treatment options) as an issue of uncertainty and less on personal uncertainty or "the unknown impact on personal goals, future wellbeing, or relationships" [61]. It is suggested that healthcare professionals take account of the type of uncertainty that is shaped by patients' personal histories and broader circumstances, and that training in this field, reflective discussions, and supervision might help them better support their patients [16, 59, 62]. Medical curricula could aim to cultivate a tolerance of uncertainty, by embracing the gray-scale space of medicine (e.g. eliminating multiple-choice assessments) and presenting uncertainty as a surmountable challenge rather than as a threat [59]. Reflective discussions could be implemented in (multidisciplinary) team meetings and patients could be provided with more (or even other than) medical information aimed at reducing (scientific) uncertainty [63]. By translating research findings into information material or connecting patients with experts by experience, they can learn from others.

## Limitations and strengths

There is no other research up to date that focuses on the phenomenon of uncertainty as experienced by people with RRMS at an early stage. The study highlights the first few months after the initial diagnosis as a period of heightened senses. Participants were able to share their experiences of events as they had just unfolded. Although interviews were directed at sharing experiences as though they were happening at that moment, a certain degree of reflection seems unavoidable. To further minimize a degree of reflection, future research could try to achieve even earlier inclusion, possibly prior to diagnosis (but this could be problematic in both a practical and ethical way), or adopt another method of data collection. Furthermore, it would be interesting to learn if and how the experience of uncertainty changes over time through a longitudinal study. This could inform healthcare practitioners at different stages of their patient's disease trajectory.

The distribution of gender and age of the participants reflects how MS mostly affects women at a young age. The majority of participants was recruited by healthcare practitioners who might have selected them on additional characteristics such as ability and eagerness to talk about their experiences, willingness to cooperate with scientific research, struggles with coming to terms with the diagnosis, et cetera. Some neurologists even expressed the expectation that certain patients would benefit from an interview. It is possible that participants experienced either more or less uncertainty than the 'average' person with MS. This could have been aggravated by the explicit focus of the study and interviews on uncertainty, suggesting there is something to be uncertain about. At the same time, the explicit focus on uncertainty is what distinguishes this study from other studies on the experiences of people with MS. Moreover, a phenomenological study is not concerned with measuring uncertainty and its influences, but tries to reveal a phenomenon as we experience it. The results should, however, be viewed in its context: a specific group of people at an early stage of a certain chronic illness in the Netherlands. In contrast to quantitative research, the findings of a qualitative, phenomenological study are not generalizable, but can be transferable to other settings [39, p262].

## Conclusions

This study explored the lived experience of uncertainty for people recently diagnosed with RRMS. It can be described as a stumbling motion across the liminal space between hope and

grief while dealing with oscillating feelings of unrest concerning the body, self, and others. Participants are overcome with an excessive focus on bodily changes, engage in a lonely and continuous pursuit of clarification, and attempt to define the value of bodily know-how compared to medical understanding. They need to find and come to terms with a new path in life, dealing with doubts, and making difficult choices. Uncertainty is also shaped by changing social relations. Others can heighten or relieve feelings of uncertainty. Not knowing what someone else is feeling or how someone else will respond, adds another layer to existing uncertainties. In moments of strong emotions or nagging worry, participants create their own logic that is not always logical, may contain inconsistencies, and sometimes only partially acknowledges or completely evades certain realities.

Adding to existing qualitative and phenomenological research into MS, and theories on uncertainty, this study portrays uncertainty as a multifaceted experience, directed at the past, present, and future. It derives from both internal and external stimuli, and evokes positive and negative emotional, cognitive, and behavioral responses in a body that is part of a complex, interwoven web of others, practices, and institutional systems. The interconnected nature of the multiple facets of uncertainty is not represented in the design of these of other studies, and could be embraced by moving away from the dichotomy between people with MS on one hand and people who live with a person with MS on the other. Even though the study highlights the first few months after the initial diagnosis, a longitudinal study could show if and how uncertainty changes over time. Although bounded to a specific context, the results could be transferrable to another setting.

The findings imply the need for a continuous attunement of healthcare practitioners to the expectations, fears, avoidance techniques, and other uniquely personal circumstances of the people with MS in front of them. Acknowledging the person with an illness as more than a diseased body, part of a complex, interwoven web of others, practices, and institutional systems, could help shift the focus in healthcare from scientific uncertainty to the type of uncertainty that is shaped by patients' personal histories and broader circumstances. Training in this field, reflective discussions, and supervision might help healthcare professionals better support their patients.

## Supporting information

**S1 Appendix. Interview guide.**
(PDF)

## Acknowledgments

The authors wish to express their profound gratitude to all the people with MS who participated in this study. We greatly appreciate their willingness to participate and openess in sharing their experiences. Secondly, we thank the neurologists (Bob van Oosten, Zoé van Kempen, Désirée Zemel, Erwin Hoogervorst and Edo Arnoldus), specialized MS nurses (Maud van Dongen, Lisette Trommelen-Verharen, Annette Hendriks-van Aalzum, Carla Kloet and Yvonne van Vuuren-Bakker), and other hospital staff (Suze Roodenburg-Kooij, Celine Schippers, Jose Broekhuijzen and José de Bont-Stikkelbroeck) that assisted with setting up the research in their institutes and recruiting patients by informing them about this study and asking them to participate. Thirdly, we are very appreciative of Nadine de Hoog who played a vital role as well in approaching and including the first participants. Fourthly, we are greatful to Herma Beuving for a close reading of a first draft of this article. Last but not least the authors

extend a special thanks to Carlo Leget for his close involvement in the first stages of this project, his critical thinking, and creative ideas.

## Author Contributions

**Conceptualization:** Eva C. van Reenen, Inge A. M. van Nistelrooij, Leo H. Visser, Alistair R. Niemeijer.

**Formal analysis:** Eva C. van Reenen, Inge A. M. van Nistelrooij, Leo H. Visser, Alistair R. Niemeijer.

**Funding acquisition:** Eva C. van Reenen, Leo H. Visser.

**Investigation:** Eva C. van Reenen.

**Methodology:** Eva C. van Reenen.

**Project administration:** Eva C. van Reenen, Janet W. K. de Beukelaar, Stephan T. F. M. Frequin.

**Resources:** Janet W. K. de Beukelaar, Stephan T. F. M. Frequin.

**Supervision:** Inge A. M. van Nistelrooij, Leo H. Visser, Alistair R. Niemeijer.

**Validation:** Inge A. M. van Nistelrooij, Leo H. Visser, Janet W. K. de Beukelaar, Stephan T. F. M. Frequin, Alistair R. Niemeijer.

**Visualization:** Eva C. van Reenen.

**Writing – original draft:** Eva C. van Reenen.

**Writing – review & editing:** Inge A. M. van Nistelrooij, Leo H. Visser, Janet W. K. de Beukelaar, Stephan T. F. M. Frequin, Alistair R. Niemeijer.

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
