## [Decision Letter · Decision Letter 0]

1 Oct 2024

PONE-D-24-03539The liminal space between hope and grief: the phenomenon of uncertainty as experienced by people living with relapsing-remitting multiple sclerosisPLOS ONE

Dear Dr. van Reenen,

Thank you for submitting your manuscript to PLOS ONE. After careful consideration, we feel that it has merit but does not fully meet PLOS ONE’s publication criteria as it currently stands. Therefore, we invite you to submit a revised version of the manuscript that addresses the points raised during the review process.

Please see the comments below.

We look forward to receiving your revised manuscript.

Kind regards,

Alejandro Botero Carvajal, MD

Academic Editor

PLOS ONE

Journal Requirements:

2. In the online submission form, you indicated that, 

data cannot be shared publicly due to potentially identifying information and lack of participant consent for sharing pseudonymized data. Data excerpts and other descriptions are presented in the article. Additional data may be available upon request from the corresponding author.

3. Your abstract cannot contain citations. Please only include citations in the body text of the manuscript, and ensure that they remain in ascending numerical order on first mention.

Reviewers' comments:

Reviewer's Responses to Questions

**Comments to the Author**

1. Is the manuscript technically sound, and do the data support the conclusions?

Reviewer #1: Yes

Reviewer #2: Partly

Reviewer #3: Yes

2. Has the statistical analysis been performed appropriately and rigorously? 

Reviewer #1: N/A

Reviewer #2: N/A

Reviewer #3: Yes

3. Have the authors made all data underlying the findings in their manuscript fully available?

Reviewer #1: Yes

Reviewer #2: Yes

Reviewer #3: Yes

4. Is the manuscript presented in an intelligible fashion and written in standard English?

Reviewer #1: Yes

Reviewer #2: No

Reviewer #3: Yes

5. Review Comments to the Author

Reviewer #1: * General aspects

As far as originality is concerned, the article can be considered to make a new and relevant contribution to this field of study by proposing to understand the experience of uncertainty lived by people recently diagnosed with multiple sclerosis. The text is coherent, but excessively long. This contributes to an uneasy reading within and between the sections of the article.

The uncertainty experienced by people with multiple sclerosis is a complex subject and deserves special attention from the academic community. We appreciate the opportunity to review this article, as it is indeed a very relevant topic.

The manuscript is clearly and coherently written and the topic is very important, especially if we consider that societies are ageing rapidly and that it is essential to pay attention to the aspects that people value when faced with a medical diagnosis and the inherent uncertainty (especially in pathologies such as multiple sclerosis). Here are some aspects that could strengthen the arguments presented and improve the article:

*Title:

It is coherent with the content presented, clear and objective. It arouses the reader's interest.

* Summary:

The abstract is clear and well organized, following the structure of the article. However, the section on results could be improved by presenting the main results of the study more clearly and objectively. The way this section is written is a little vague and does not allow a clear link to be established with what is presented in the conclusion section. In turn, it is too focused on health professionals (rather than the experience of existential uncertainty by people with multiple sclerosis). It might also be useful to include more clearly the contribution/implications of the study for research in the field.

*Introduction:

The introduction adequately presents the context and importance of the study. It is clearly written, but perhaps a little briefer than is desirable in terms of the existing research on the subject. In other words, the introduction gives a reasonably clear account of the implications of uncertainty related to medical diagnosis on patients' well-being and quality of life, but it is more vague when it comes to research into the specific scope of this study - the uncertainty experienced by people with multiple sclerosis. The article would advance significantly if the authors presented a more complete overview of the relevant research in this field, in order to make the research problem more explicit. In other words, the article would benefit from a more detailed analysis of recent research, and the results obtained, in this specific field of study.

That said, it is important that the introduction situates the existing knowledge on the topic under study, helping to formulate the hypotheses/arguments that will be developed in the article.

It is also necessary to make a link between previous work and this article. For example, if the following is stated: “A phenomenological study of the lifeworld of people with MS is not a new venture”, it would be interesting to explore further why the approach adopted is innovative. In other words, it could be useful to clarify the originality and relevance of the study, since the innovative nature of the study is not entirely clear.

The theoretical approach used seems adequate for reading and interpreting the results obtained and the introduction ends with a presentation of the study's objective - which is very appropriate.

*Methodology:

The authors adequately describe the methodology used, however this section is quite extensive. The article would benefit from a shorter, less detailed section. For example, the description of the study design and the participant selection process could be more succinct. Omitting the names of the hospitals might also be appropriate in order to guarantee the anonymity of the participants.

Given the nature of the study, the authors seem uncomfortable with “pre-defined” designations, as they state: “For practical reasons, this study upholds these concepts while acknowledging their problematic nature”. It is therefore suggested that a structure more in keeping with a phenomenological study be adopted. Furthermore, it doesn't seem to make much sense to mention the tasks that were done (or not) by the first author of the study (or by others). Similarly, the ethical considerations should be shorter (for example, it doesn't seem relevant to mention that a participant became emotional during the interview, or what the researcher's behavior was at that moment). Thus, the section on the methodology used should describe the methods and techniques used clearly and objectively, and non-essential aspects should be eliminated or summarized.

*Results

The results section begins with a sentence indicating the bibliographic reference and page. I don't understand why this is: “In this section, the essence of the researched phenomenon and its constituents will be described [31 p.255].”

This is followed by a paragraph in which the authors frame what will be presented next, but it is not clear whether these ideas derive specifically from the interviews conducted. I don't think a framework of this length is justified. If it is to be kept, it should be shorter.

The results section is around 17 pages long and that seems too long to me. As has been suggested throughout the article, we recommend reducing this section. The content is very interesting, but it could clearly focus on the essentials and present the main results of the study carried out - illustrated, if possible, with extracts from the interviews conducted. This would make the reading more objective, richer and more interesting, as the transcripts are adequate and useful for understanding the participants' perspective.

* Discussion

The discussion begins with a summary of the main results, which is appropriate. Nevertheless, some repetition could be eliminated. There is room for a more in-depth discussion about the uncertainty of people diagnosed with Multiple Sclerosis and the role of health professionals.

What follows is a set of arguments that would make more sense in the introduction of the article rather than in the discussion. At this point, one would expect a more in-depth critical analysis of the main results of the study. What often happens is that one of the results obtained is presented and compared with one or more bibliographic sources, but there is a lot of information about these studies (e.g. how they were conducted, limitations, etc.), which makes reading difficult and not very fluid.

It is therefore suggested that the theoretical and empirical framework be more robust in the introduction so that the discussion can adopt a more analytical and critical lens on the results obtained. There is also a need for greater fluidity in the reading of this section (and the whole article) which could be achieved by significantly reducing the size of the article.

With regard to the section with recommendations for healthcare practices, we suggest eliminating the first few paragraphs which repeat what has already been mentioned above and keeping only the authors' recommendations based on the research carried out (duly supported by theory and research).

As far as the main limitations of the study are concerned, the constitution of the sample may be a weakness (especially the fact that most of the participants were recruited by health professionals) as it is not a study with a random sample. As the authors point out, this has implications for the generalizability of the results. It might also be useful to analyze the cultural influence on the experience of uncertainty, since we are analyzing a specific country.

It is also essential to make a clear statement about the contribution and/or implications of the study in relation to existing knowledge. What, in fact, are the main conclusions of the study and what implications does it have for the advancement of knowledge in this field? What implications does a study, albeit somewhat exploratory, on individual perceptions (of people with multiple sclerosis) of uncertainty have?

*Conclusion

Based on the results of the study, it can be concluded that uncertainty is a multifaceted experience and that people newly diagnosed with multiple sclerosis have to deal with various levels of uncertainty. In this context, it is up to healthcare professionals to support these patients - not only with clear and objective medical information, but also with a greater understanding of their doubts, difficulties and uncertainties. Given that these are the main conclusions of the study, what implications do they have for the advancement of knowledge in this field?

It is also very important to elaborate a little more on what is recommended in the final paragraph (“Training in this field, reflective discussions, and supervision might help healthcare professionals better support their patients.”).

Reviewer #2: The topic chosen by the authors is worth investigating. The uncertainty and unpredictability associated with managing conditions such as relapsing-remitting Multiple Sclerosis (RRMS) are most often inadequately addressed, leading to misunderstandings about the condition.

Patients with RRMS frequently experience unpredictable or unfamiliar symptoms, which can cause uncertainty and challenges in their daily lives. Such unpredictability and uncertainty may also lead healthcare providers to overlook patients' experiences of the condition. They may also overlook essential aspects of care. However, different aspects of uncertainty explored in this study can help enhance awareness among healthcare practitioners and patients. This can, in turn, result in developing strategies that empower individuals to navigate their condition with greater confidence, thereby improving patient care.

While the manuscript addresses an important topic, it is currently somewhat difficult to follow. The authors have used several technical terms and jargon, which could make it challenging for the broader scientific community to fully understand. It is essential that the manuscript be presented in a clear format to ensure the research is understood, and its implications are fully recognised. Examples include ‘attunement’, ‘uniquely’, ‘antecedents’, ‘irreconcilable’ etc

Line 117, the statement "[uncertainty] is a lack of knowledge of which the individual is aware" is somewhat unclear. Please improve clarity.

The final paragraph of the introduction regarding phenomenology may be shifted to the methods section.

Study Design

The study design is too lengthy and may cause confusion. Please revise it for clarity and focus.

“The gap between on the hand hand a descriptive and on the other hand an interpretive approach was closed by Dahlberg 216 and Dahlberg’s third way.” Does not read well.

“Second, a medical student who worked as an intern in both the 231 Elisabeth-TweeSteden Hospital and on this research project, actively searched 232 patient files and reached out to possible candidates after consulting with their 233 physician.”

• Did the candidates/participants know that their information was being accessed for research purposes? And how was patient confidentiality maintained when accessing their files?

Line 257, Data Collection

• First paragraph: The authors introduce the concept of data within the framework of phenomenology; however, this section could benefit from being simplified or even removed. Supporting references could enhance its relevance.

Please remove duplicates

• e.g. line 241-42 and line 267-68 cover the same content

Line 284: follow-up questions or probing questions?

Please be consistent in using either 'participant' or 'candidate.'

Line 287-288: This implies that the interview guide has not been the core component of the interview process or perhaps a secondary resource. Please clarify.

The data analysis and Ethics sections could benefit from further refinement, as they currently appear lengthy. A more concise presentation will enhance clarity and strengthen the manuscript's overall impact. It might be helpful to focus on key points while ensuring that essential details are still communicated effectively.

Results Section

There is a lot of information in the content, which shows the richness of the result. However, it might be easier to understand if it were reviewed and shortened. For instance, the section on the "search for definition" could be revised by highlighting a few key issues that directly address the main point while summarising the remaining content. This would enhance focus and readability.

Given the points mentioned above, I recommend revising the manuscript before resubmitting it for another review. Therefore, I will pause my review of the results and discussion sections at this time. Thank you

Reviewer #3: Thank you for a well-prepared manuscript which I had the joy to read. The manuscript addresses a significant and often overlooked aspect of living with relapsing-remitting multiple sclerosis (RRMS). I have read the manuscript carefully. Generally, your manuscript is well-written and well-structured. To improve your manuscript, I have a few comments for you.

Introduction: The introduction is well-written and describes the previous research on the topic and defines the scope of the research well. However, I think it would benefit from a more explicit specification of the research question.

Methods: The methods are described very well and in sufficient detail. However, the study design section is somewhat lengthy and a bit confusing. It would be clearer if the authors focused on describing what the study is, rather than what it is not.

Recruitment: The anonymity of informants should be observed carefully.

The relevance of “Chosen medication” and “medication and usage” is unclear. It might be more pertinent to include information about social support instead.

There is an overlap between Table 1 and Table 2; choosing one would streamline the presentation.

The phrase “As described previously” leads to repetition of information and should be omitted.

The statement “interviews that were conducted immediately after inclusion” needs clarification, as it does not align with the description on page 16, lines 337-338, where it states that “all participants were briefed by the interviewer (EvR), first by email or phone and second in person, preceding the actual interview.” Please elaborate on what “immediately” means.

What is the significance of informing about how long the interviews lasted?

How do the interviewers relate to the informants?

The use of journals to “capture non-verbal communication, atmosphere, and other relevant observations” should be explained in terms of how these observations were utilized.

Results: The results are well analyzed and written. However, the subtheme “Painful memory of the past” does not accurately describe the content, which seems to focus more on the participants’ difficulties in adjusting to their new situation in everyday life.

Discussion: The heading “Discussion” competes with the subheading “Reflections on the findings.” Please specify whether this section is a discussion or a reflection, and introduce what is to be discussed or reflected on more precisely.

Recommendations for Healthcare Practice: This paragraph is quite long. I suggest you to focus on the implications for practice derived from the findings, rather than referencing other works.

Limitations and Strengths: Please discuss whether it is a limitation or a strength that the focus on the experience of uncertainty in people with RRMS is at such an early stage. Consider discussing the pros and cons of this. Additionally, why is future research at an earlier stage recommended? Could a longitudinal study be explicitly recommended for future research?

Thank you for considering my feedback. I believe these revisions will enhance the clarity and impact of the manuscript.

6. PLOS authors have the option to publish the peer review history of their article (what does this mean?). If published, this will include your full peer review and any attached files.

Reviewer #1: No

Reviewer #2: No

Reviewer #3: **Yes: **Lise Sæstad Beyene

---

## [Author Response · Author response to Decision Letter 0]

15 Nov 2024

Dear dr. Botero Carvajal,

Thank you for the decision and detailed response on our submitted manuscript (PONE-D-24-03539). We also appreciate the recommendations for amendments and clarifications by the three reviewers. We have thoroughly revised our manuscript, which you will find attached. Below I will respond to each of the comments separately and as detailed as possible. 

1. We apologize for any errors in the formatting of our manuscript, including the file names. We have carefully checked the style requirements again and have attempted to fully comply with them.

2. Public deposition would indeed breach compliance with the protocol approved by our research ethic board. In contrast to the data excerpts and other descriptions presented in the manuscript, the full interview transcripts contain very personal and detailed private information of the participants. We are therefore concerned about the traceability of the data and the perceived privacy by our participants. We kindly ask for your understanding and exemption and if not, hope you will reach out to us to jointly seek a fitting solution.

3. The number 17 refers to the number of participants, not to a citation. We have now changed “17” into “Seventeen” in the “Abstract” section. 

4. In the manuscript it now states: 

Prior to the start of this study, the Medical Ethical Review Committee Brabant (The Netherlands) established that the Dutch Medical Research Involving Human Subjects Act (WMO) did not apply (NW2020-65). In addition, the Ethical Review Committee of the University of Humanistic Studies Utrecht (The Netherlands) concluded that the rights, safety, and well-being of participants was guaranteed (2020.6). Furthermore, approval was obtained from the Review Boards of the four participating hospitals: the Elisabeth-TweeSteden Hospital Tilburg (L1135.2020), St. Antonius Hospital Utrecht (Z21.016), VU Medical Centre Amsterdam (2020.0715), and Albert Schweitzer Hospital Dordrecht (2021.044). Pseudonimity was ensured by assigning a number to all participants and removing as much identifying information from the transcripts as possible. Data were stored in a secured project file and access was restricted to members of the research team. All participants gave written informed consent.

General aspects:

We appreciate the reviewers’ constructive comments and the recognition of the importance of exploring the topic of uncertainty. In response to the feedback, particularly concerning the length of the manuscript, we have significantly reduced the text by nearly 3000 words. This should improve readability and streamline the flow of the paper. Additionally, as suggested by reviewer #2, we have aimed to make the manuscript more accessible to a broader scientific audience, by reducing the use of technical terms and jargon. We have also provided clearer explanations of the steps in our phenomenological approach.

Abstract:

As suggested by reviewer #1, we have revised the results section to present the results more clearly whilst matching the structure of essence and constituents of a phenomenon. Additionally, the conclusion now briefly highlights the study’s contribution to research in the field and is less elaborate on the implications for health professionals.

Introduction:

In contrast to the rest of the article, reviewer #1 considers the “Introduction” to be briefer than desirable in terms of the existing research on the subject. The other reviewers provided more specific comments. Before detailing the changes made in response, we would like to make the following general statement. Some of the suggestions offered by the reviewers, regarding the introduction and other elements of the manuscript, do not always fully match a phenomenological research design. Consequently, we have incorporated their feedback where appropriate. 

 As the “Introduction” is concerned, reviewer #1 recommends a more detailed analysis of recent research in this specific field of study. Phenomenological research, however, often avoids extensive literature reviews in the early stages of a project. In order to investigate a lived experience as openly as possible, the introduction mostly identifies gaps in the literature and contains little theorization. This is why most theory is introduced and conversed with in the “Discussion”, instead of in the “Introduction”.

 We have, however, made some changes in this section whilst keeping in mind the general comments of the reviewers on the length of the article. These changes focus on strengthening the research problem (reviewer #1) and clarifying the research question (reviewer #3). We have removed lines 115-117 as the attempted clarification of the cited definition of uncertainty in the previous sentence, was actually unclear to reviewer #2, and may be superfluous. We decided not to move the final paragraph of the “Introduction” to the “Methods” section, as suggested by reviewer #2, since this text contains the study’s objective and is an appropriate ending to the “Introduction” according to reviewer #1. 

Methods:

All three reviewers noted the length, excessive detail, and complexity of the “Study design” section. We have significantly shortened this section, providing a focused description of the two adopted phenomenological research methods. Additionally, the reviewers emphasized the need to safeguard participant anonymity. In response, we made several adjustments under “Recruitment and participants” to meet their concerns: the role of the intern is described differently to prevent any misunderstanding on how patient files were employed, and table 2 is removed. The names of the participating hospitals have been omitted here, but are still mentioned under “Ethical considerations” to meet the requirements of the ethics statement. Moreover, the names of the participating hospitals would be traceable nonetheless through approval numbers and author affiliations. Furthermore, a comment by reviewer #3 pointed us to the relevance of mentioning medication for this article. Medication usage was indeed one of the inclusion criteria of the larger prospective study, but the analysis of this aspect is reported on in a separate article. We therefore stated this more clearly and removed all references to medication from the table with participant characteristics. 

 Third, the introduction of the concept of data within the framework of phenomenology under “Data collection” raised questions as well. Following reviewer #2’s advice, we have removed the paragraph altogether; we assume the practical application of this and other external concepts within phenomenological research is implied. Despite the doubts expressed by reviewer #3 on the significance of informing the reader about how long the interviews lasted, we decided to disclose these details nonetheless. Relatively short interviews would not be fitting within a phenomenological approach. We have, however, tried to clarify the role of the use of an interview guide and journal, as reviewer #3 suggested. Furthermore, we refined the description of the “Data analysis”, as proposed by reviewer #2. Fifth, the “Ethical considerations” were significantly shortened, as recommended by reviewers #1 and #2, by focusing on key points. 

 Finally, we addressed repetition (reviewers #1 and #2), ensured a consistent use of “participants” over “candidates” (reviewer #2), and reduced unnecessary details about task assignments (reviewer #1). 

Results:

To reviewer #1, the purpose and content of the beginning of the “Results” section is not clear. Starting with a description of the “essence” of a phenomenon, followed by its constituents, is a trademark of the Reflective Lifeworld Approach, which is explained in the “Methods” section. We have, however, made some revisions in this first paragraph, to better guide the reader through the beginning of this section.

 Regarding the length of the “Results” section, two of the three reviewers raised concerns. We agree with the reviewers that a text should be focused and readable, but also aim to uphold the principles of a phenomenological approach. Reviewer #1 suggested that a structure more consistent with a phenomenological study should be adopted, which we believe supports retaining a certain level of detail in the results. Presenting only “the essentials” (reviewer #1) or “a few key issues” (reviewer #2), would diminish the “richness of the results” that reviewer #2 appreciated. It is through thick descriptions that the reader is invited to vicariously experience the uncertainty of people with RRMS. We have, however, critically reviewed this section, removing repetitive elements and excessive examples. Combined with the overall reduction in the manuscript’s length, we feel this enhances the readability of this section. Furthermore, we have revised the subheading “Painful memory of the past” to better reflect the content, as suggested by reviewer #3.

Discussion:

The “Discussion” section has been substantially modified, starting with a clearer introduction to the content of the following paragraphs, addressing reviewer #3’s concerns. Furthermore, to improve readability and flow (reviewer #1), we have removed extensive details about the bibliographic sources, instead focusing on a more analytical discussion of the results in relation to existing literature. In line with a phenomenological research approach, we maintain that much of the literature introduced here is better suited to the discussion than the introduction (reviewer #1). This allows for an open research design and question instead of imposing existing theories, which follow from the results instead of the other way around.

Recommendations for healthcare practice:

We have followed the advice of both reviewer #1 and #3 to shorten this section by eliminating the first paragraph, which summarized the most important findings for healthcare practice. The section now focuses solely on the practice implications of our findings. While reviewer #3 suggests minimizing references in this section, we align with reviewer #1’s perspective that recommendations should be supported by relevant theory and research. This ensures that our suggestions are well-grounded and credible.

Limitations and strengths:

As reviewer #1 suggested, we have clarified the value of an exploratory study on individual perceptions of uncertainty. The main conclusions and their implications for advancing knowledge in this field are further elaborated on in the “Conclusion”. We have explicitly highlighted the contribution of exploring uncertainty in people with RRMS shortly after diagnosis. In response to reviewer #3, we discuss the early-stage focus on uncertainty as primarily a strength, emphasizing its relevance for improving early care and support. Additionally, we have considered the pros and cons of this approach and suggested that future research could benefit from a longitudinal design to capture the evolving nature of uncertainty over time.

Conclusions:

Only reviewer #1 provided feedback on the “Conclusions” section. In response, we have added text discussing the implications of our findings for the advancement of knowledge in this field. Additionally, as reviewer #1 suggested, we have elaborated on the final paragraph for healthcare professionals. However, this elaboration has been included under “Recommendations for healthcare practice” to avoid repetition in the “Conclusions” section.

Thank you for receiving our revised manuscript. We appreciate your time and look forward to your response.

Kind regards, on behalf of all authors,

Eva van Reenen

---

## [Decision Letter · Decision Letter 1]

27 Nov 2024

The liminal space between hope and grief: the phenomenon of uncertainty as experienced by people living with relapsing-remitting multiple sclerosis

PONE-D-24-03539R1

Dear Dr. van Reenen,

We’re pleased to inform you that your manuscript has been judged scientifically suitable for publication and will be formally accepted for publication once it meets all outstanding technical requirements.

Kind regards,

Alejandro Botero Carvajal, MD

Academic Editor

PLOS ONE

Additional Editor Comments (optional):

Reviewers' comments:

Reviewer's Responses to Questions

**Comments to the Author**

1. If the authors have adequately addressed your comments raised in a previous round of review and you feel that this manuscript is now acceptable for publication, you may indicate that here to bypass the “Comments to the Author” section, enter your conflict of interest statement in the “Confidential to Editor” section, and submit your "Accept" recommendation.

Reviewer #1: All comments have been addressed

Reviewer #3: All comments have been addressed

2. Is the manuscript technically sound, and do the data support the conclusions?

Reviewer #1: Yes

Reviewer #3: Yes

3. Has the statistical analysis been performed appropriately and rigorously? 

Reviewer #1: N/A

Reviewer #3: N/A

4. Have the authors made all data underlying the findings in their manuscript fully available?

Reviewer #1: (No Response)

Reviewer #3: Yes

5. Is the manuscript presented in an intelligible fashion and written in standard English?

Reviewer #1: Yes

Reviewer #3: Yes

6. Review Comments to the Author

Reviewer #1: (No Response)

Reviewer #3: Thank you for the opportunity to review your revised manuscript. I acknowledge the significant progress you have made since the initial submission. The improvements enhance the clarity and depth of your research. I recommend this manuscript to be accepted for publication. It contributes valuable insights to the field and advances our understanding of the topic.

7. PLOS authors have the option to publish the peer review history of their article (what does this mean?). If published, this will include your full peer review and any attached files.

Reviewer #1: No

Reviewer #3: No

---

## [Editor Report · Acceptance letter]

17 Jan 2025

PONE-D-24-03539R1 

PLOS ONE

Dear Dr. van Reenen, 

I'm pleased to inform you that your manuscript has been deemed suitable for publication in PLOS ONE. Congratulations! Your manuscript is now being handed over to our production team.

Kind regards, 

on behalf of

Dr. Alejandro Botero Carvajal 

Academic Editor

PLOS ONE